

# Pyrolyzed and unpyrolyzed residues enhance maize yield under varying rates of application and fertilization regimes

Owais Ali Wani[1], Farida Akhter[1], Shamal Shasang Kumar[1], Raihana Habib Kanth[2], Zahoor Ahmed Dar[3], Subhash Babu[4], Nazir Hussain[5], Syed Sheraz Mahdi[2], Abed Alataway[6], Ahmed Z. Dewidar[6,7] and Mohamed A. Mattar[6,7]

[1] Division of Soil Science and Agricultural Chemistry, Faculty of Agriculture, Sher-e-Kashmir University of Agricultural Sciences and Technology of Kashmir, Kashmir, India
[2] Division of Agronomy, Faculty of Agriculture, Sher-e-Kashmir University of Agricultural Sciences and Technology of Kashmir, Kashmir, India
[3] DARS- Rangreth, Rangreth, India
[4] Division of Agronomy, ICAR - Indian Agricultural Research Institute, New Delhi, India
[5] KVK Kargil, Kargil, India
[6] Prince Sultan Bin Abdulaziz International Prize for Water Chair, Prince Sultan Institute for Environmental, Water and Desert Research, King Saud University, Riyadh, Saudi Arabia
[7] Department of Agricultural Engineering, College of Food and Agriculture Sciences, King Saud University, Riyadh, Saudi Arabia

Corresponding authors
Owais Ali Wani,
owaisaliwani@gmail.com
Mohamed A. Mattar,
mmattar@ksu.edu.sa

## ABSTRACT

Biochar is increasingly gaining popularity due to its extensive recommendation as a potential solution for addressing the concerns of food security and climate change in agroecosystems, with biochar application for increased carbon sequestration, enhanced soil fertility, improved soil health, and increased crop yield and quality. There have been multiple studies on crop yield utilizing various biochar types and application amounts; however, none have focused on the influence of diverse biochar types at various pyrolysis temperatures with different application amounts and the integration of fertilizer regimes in maize crops. Therefore, a two-year factorial field experiment was designed in a temperate Himalayan region of India (THRI) to evaluate the residual effect of different biochar on maize yield under different pyrolysis temperatures, various application rates and fertilizer regimes. The study included three factors *viz.*, amendment type (factor 1), rate of application (factor 2) and fertilizer regime (factor 3). Amendment type included 7 treatments: No biochar- control (A1), apple biochar @ 400 °C pyrolysis temperature (A2), apple biochar @ 600 °C pyrolysis temperature (A3), apple residue biomass (A4), dal weed biochar @ 400 °C pyrolysis temperature (A5), dal weed biochar @ 600 °C pyrolysis temperatures (A6), and dal weed residue biomass (A7). The rate of application included 3 levels: Low (L- 1 t ha$^{-1}$), medium (M- 2 t ha$^{-1}$), and high (H- 3 t ha$^{-1}$). At the same time, the fertilizer regimes included 2 treatments: No fertilizer (N) and recommended dose of fertilizer (F). The results revealed that among the various amendment type, rate of application and fertilizer regimes, the A3 amendment, H rate of application and F fertilizer regime gave the best maize growth and productivity outcome. Results revealed that among the different pyrolyzed residues used, the A3 amendment had the highest plant height (293.87 cm), most kernels cob$^{-1}$ (535.75), highest soil plant analysis development (SPAD) value (58.10), greatest cob length (27.36 cm), maximum cob girth (18.18 cm), highest grain cob yield (1.40 Mg
ha$^{-1}$), highest grain yield (4.78 Mg ha$^{-1}$), higher test weight (305.42 gm), and highest stover yield (2.50 Mg ha$^{-1}$). The maximum dry weight in maize and the number of cobs plant$^{-1}$ were recorded with amendments A4 (14.11 Mg ha$^{-1}$) and A6 (1.77), respectively. The comparatively 2$^{nd}$ year of biochar application than the 1$^{st}$ year, the H level of the rate of application than the L rate and the application and integration of the recommended dose of fertilizer in maize results in significantly higher values of growth and productivity in maize. Overall, these findings suggest that the apple biochar @ 600 °C pyrolysis temperature (A3) at a high application rate with the addition of the recommended dose of fertilizer is the optimal biochar for enhancing the growth and productivity of maize in the THRI.

# INTRODUCTION

Agricultural productivity is perpetually constrained by a periodic decline in soil quality and poor nutrient use efficiencies, resulting in food insecurity (*Naorem et al., 2023*). Challenges such as climate change, population growth, and urbanization strain agroecosystems, necessitating a review of their form and function to address issues like poor nutrient supply, utilization, recycling, and water efficiency (*Kumar, Mahale & Patil, 2020*; *Lal, 2013*). Recycling organic nutrients back into the soil could be one effective strategy for preserving soil organic matter (SOM), which often enhances both the physicochemical properties of the soil (*Dungait et al., 2012*). Agricultural crop residues, encompassing animal, household, and industrial waste, represent significant forms of organic matter for soil incorporation (*Sarkar et al., 2020*). However, careful selection of organic material sources is necessary due to variations in quality and potential contaminants, as some sources may adversely affect soil health (*Jones & Healey, 2010*). Biochar, also known as "biomass-derived black carbon (C)" or "charcoal", stands out as a soil additive with unique properties capable of enhancing soil fertility and crop productivity (*Akhtar et al., 2014*). It serves as a long-term C sink, often referred to as "agrichar" due to its potential in sustainable soil management and addressing climate change concerns (*Abewa et al., 2014*; *Lehmann, Gaunt & Rondon, 2006*; *Yao et al., 2012*). Produced through the pyrolysis of agricultural waste at high temperatures and anoxic conditions, biochar is rich in C and contains essential nutrients like hydrogen (H), oxygen (O$_2$), magnesium (Mg), nitrogen (N), phosphorous (P), and potassium (K), which contribute to increased crop production (*Alkharabsheh et al., 2021*; *Smith, 2016*). In the past two decades, biochar has gained significant recognition in agriculture for its diverse applications, including C sequestration, bioremediation, soil fertility enhancement, wastewater treatment, and overall environmental management (*Diatta et al., 2020*). *Bonanomi et al. (2017)*, *Rawat, Saxena & Sanwal (2019)* and *Chan et al. (2008)* enhanced crop yield and nutrient uptake are typically linked to direct nutrient contributions from applied biochar, which contains diverse nutrients. However, crop interactions with biochar vary based on biochar type,

plant species, and soil type (*Hussain, Garg & Ravi, 2020*; *Borchard et al., 2019*; *Cayuela et al., 2014*; *He et al., 2017*; *Abd Elwahed et al., 2019*; *Zee, Nelson & Newdigger, 2017*; *Trupiano et al., 2017*; *Choudhary et al., 2021*; *Satriawan & Handayanto, 2015*).

Biochar, often termed the "black gold" of agriculture, boasts a highly diverse composition containing stable and labile components. With increasing pyrolysis temperature, the proportion of aromatic C in biochar rises due to heightened volatile matter loss and the conversion of alkyl and O-alkyl C to aryl C (*Demirbas, 2004*). Each biochar particle is widely assumed to include two major structural fractions: stacked crystalline graphene sheets and randomly organized amorphous aromatic structures. Their principal constituents are commonly considered to be C, volatile matter, mineral matter (ash), and moisture (*Mansoor et al., 2021*). As biochar becomes incorporated into the soil, it alters a wide array of physiochemical and biological soil properties, including C content, pH, cation exchange capacity (CEC), porosity, surface area, bulk density, water-holding capacity (WHC) and nutrient utilization efficiency (NUE) (*Seleiman et al., 2020*; *Weber & Quicker, 2018*). (*Pariyar et al., 2020*; *Inyang & Dickenson, 2015*; *Seleiman et al., 2020*; *Tomczyk, Sokołowska & Boguta, 2020*; *Kavitha et al., 2018*). The combination of feedstock type and pyrolysis conditions facilitates the production of biochars with distinct characteristics (*Mukherjee & Zimmerman, 2013*). The impact of various pyrolysis temperatures and application rates on maize yield can vary significantly depending on factors such as soil type, climate conditions, crop management practices, and the specific characteristics of the biochar produced (*Wang et al., 2015*). Different pyrolysis temperatures influence the chemical composition and physical properties of biochar, which can subsequently affect its effectiveness as a soil amendment (*Tomczyk, Sokołowska & Boguta, 2020*). Higher pyrolysis temperatures often result in biochars with higher C content, greater stability, and altered nutrient availability. These changes may influence soil nutrient dynamics, water retention capacity, microbial activity, and plant nutrient uptake, ultimately impacting maize yield (*Das, Ghosh & Avasthe, 2022*). Similarly, varying application rates of biochar can have diverse effects on maize yield. Higher application rates may enhance soil fertility, improve soil structure, increase nutrient retention, and promote microbial activity, leading to potential increases in maize yield (*Diatta et al., 2020*). However, excessively high application rates can also cause issues such as nutrient imbalance, soil compaction, and reduced water infiltration, which may negatively impact crop productivity (*Alkharabsheh et al., 2021*).

Variations between the type of biochar used leads to differences in the soil chemical properties such as pH, CEC, SOC, EC, and extractable nutrients (*Wang et al., 2013*; *Gray et al., 2014*; *Gray et al., 2014*; *Ahmad et al., 2014*; *Jeffery et al., 2015*).

Biochar application methods, despite receiving limited attention significantly impact soil health. Its placement near the soil surface in the root zone is ideal that maximizes nutrient cycling and plant uptake. Techniques include broadcasting by hand, using tractor-propelled spreaders, deep banding in the rhizosphere, and mixing with solid or liquid fertilizers (*Kapoor et al., 2022*). Biochar's diverse physicochemical qualities influence application rates, with studies showing improvements in crop yields at 5–50 tons per acre with proper nitrogen management (*Chan et al., 2008*). The intended rate, biochar supply, and soil management dictate application frequency, with benefits persisting across multiple

growing seasons due to its recalcitrance (*Steiner et al., 2007*). While biochar's positive effects on soil are believed to enhance over time (*Mansoor et al., 2021*), its combined impact with pyrolyzed and unpyrolyzed additions and fertilizer regimens on maize productivity remains unclear. Therefore, we hypothesize that varying combinations of biochar treatments, application rates, and fertilizer regimens will result in differential effects on maize productivity.

In current study we hypothesized effect of pyrolyzed and unpyrolyzed feedstock applied in various rates and with and without combination of fertilizers applied on growth and yield of maize crop in temperate ecosystem of North-western Himalayas.

## MATERIALS & METHODS

### Site description
The research took place at the Wadura site of the Sher-e-Kashmir University of Agricultural Sciences and Technology of Kashmir, situated in Sopore, Kashmir, at 34°17′N latitude and 74°33′E longitude (Fig. 1). Wadura is located at an elevation of 1,524 m above sea level and features a flat surface with good drainage. Based on meteorological data collected during the study, peak temperatures during the Kharif seasons of 2020 and 2021 ranged from 22.80 to 38.00 °C, while weekly minimum temperatures were between 8.70 and 19.30 °C. The average annual precipitation is 812 mm, with more than 80% of it attributed to western disturbances. The soil in the study area is classified as inceptisols, with a neutral pH and low availability of N and P, and medium availability of K.

### Biochar production
The biochars were produced using patented technology IOT-based multipurpose pyrolyzer cum heater cum cooker. It works on the principle of a system approach patented under patent number 202011056587. The feedstock underwent pyrolysis at two distinct temperatures: 400 °C and 600 °C, individually, for 3 h, with a gradual increase of 8 °C per minute. The treatment consisted of three levels of rate of fertilizers viz., low (L- 1 t ha$^{-1}$); medium (M- 2 t ha$^{-1}$); high (H- 3 t ha$^{-1}$).

FTIR spectroscopy offers valuable insights into the molecular composition and structural properties of biochar derived from varying pyrolysis temperatures. By subjecting biomass to different thermal conditions, biochar is formed with distinct chemical functionalities and bonding arrangements. In this study through FTIR analysis of biochar, we elucidated the spectral signatures corresponding to functional groups such as hydroxyl, carbonyl, and aromatic structures, which undergo transformations during pyrolysis. Understanding these spectral variations can illuminate the influence of pyrolysis temperature on biochar properties, aiding in the optimization of production processes for desired biochar applications (Figs. 2A, 2B).

### Experimental design, treatment details and field operation
An experiment was set up with three replications, thirty-eight treatments in a factorial randomized complete block design (F-RCBD). The treatment consisted of three factors, having 1st factor with seven levels of amendments viz., no material (A1); apple biochar

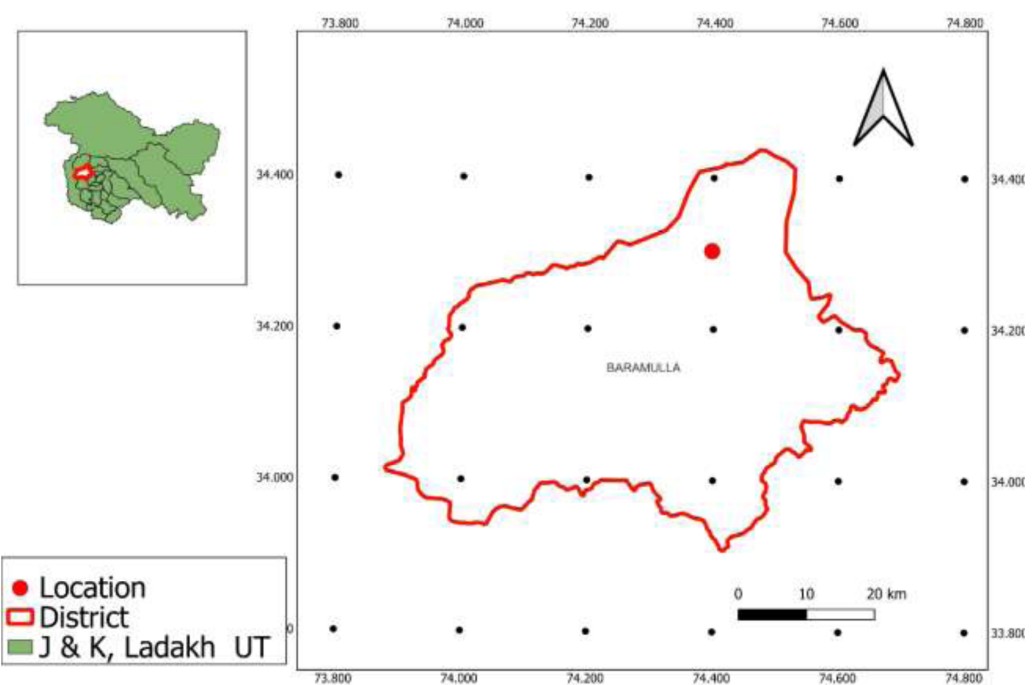

**Figure 1  Study area map of experimental location.**

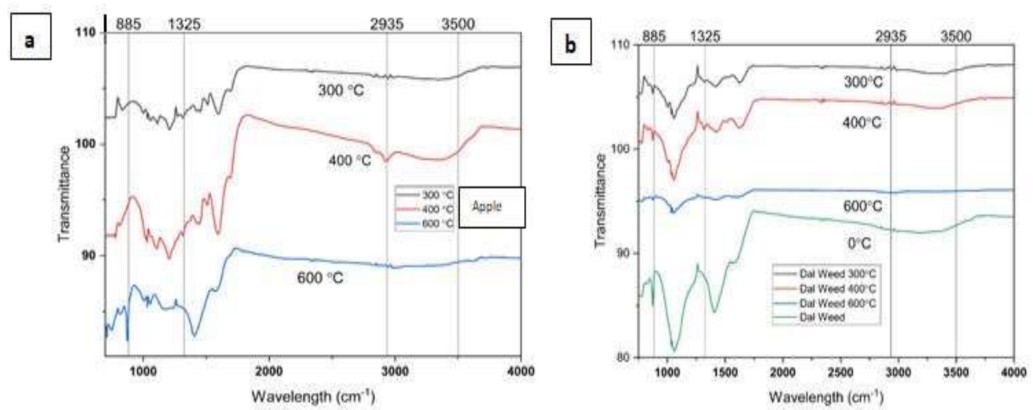

**Figure 2  FTIR spectra of Biochars produced at different pyrolysis temperatures.**

(400 °C) (A2); apple biochar (600 °C) (A3); apple residue biomass (A4); dal weed biochar (400 °C) (A5); dal weed biochar (600 °C) (A6); dal weed residue biomass (A7), 2nd factor with three levels of rate of fertilizers viz., low (L- 1 t ha$^{-1}$); medium (M- 2 t ha$^{-1}$); high (H- 3 t ha$^{-1}$) and the 3rd factor with two levels of inorganic fertilizers *viz.*, no fertilizer (N) and recommended dose of fertilizer (RDF) (F) (Table 1). The experiment consisted of forty-two treatment combinations with three replicates amounting to a total of 126 experimental units.

**Table 1  Treatment details.**

| Factor 1 (Amendments) | Factor 2 (Rate) | Factor 3 (Inorganic fertilizer) |
|---|---|---|
| 1. No material - ($A_1$) | Low - (L) | No fertilizer - (N) |
| 2. Apple biochar (400 °C ) - ($A_2$) | Medium - (M) | RDF (recommended dose of fertilizer) - (F) |
| 3. Apple biochar (600 °C ) - ($A_3$) | High - (H) | |
| 4. Apple residue biomass - ($A_4$) | | |
| 5. Dal weed biochar (400 °C ) - ($A_5$) | | |
| 6. Dal weed biochar (600 °C ) - ($A_6$) | | |
| 7. Dal weed residue biomass - ($A_7$) | | |

## Plant height (cm)

The plant height measurement process involved tracking five specifically tagged plants situated in the penultimate rows of each plot. These measurements were conducted at 15-day intervals. The recorded heights of these plants were then averaged to derive the overall plant height in centimetres. The measurement encompassed the distance from the base of the soil surface to the fully opened top leaf, offering a comprehensive assessment of the plants' vertical growth progress over time using a meter ruler.

## Dry matter accumulation (t ha$^{-1}$)

Fifteen days after Sowing, three plants were randomly chosen from the penultimate row of every plot. These plants were harvested, meticulously chopped into small pieces, and then thoroughly mixed to ensure homogeneity. Subsequently, they underwent oven drying at a temperature range of 60–65 °C until reaching a constant weight. Following this, the dry weight was recorded using an electronic balance, and the values were averaged and expressed as grams per plant (g plant$^{-1}$). Finally, these values were converted into **t ha$^{-1}$** to provide a standardized measure of biomass yield across the experimental plots.

## Number of cobs plant$^{-1}$

Before the picking process, the total number of green cobs within five randomly marked plots was carefully counted.

## Number of kernels cob$^{-1}$

In each plot, the number of kernels on five randomly selected cobs was manually counted. This meticulous counting process allowed for the determination of the average number of grains per cob. The resulting data provided valuable insights into the grain yield potential and variability across the experimental plots.

## SPAD value

The relative chlorophyll content in maize plant leaves based on light absorption involved the use of a handheld SPAD-502 chlorophyll meter.

## Cob length (cm)

The measurement of maize cob length was conducted following the harvesting of mature maize cobs. After removing any extraneous leaves or husks from five randomly selected cobs per plot, each cob was individually measured using a centimetre ruler. The measurement

starts from the base where the cob attaches to the stalk and extends to the tip of the cob. Care was taken to ensure the measuring tool was aligned straight along the length of the cob for accurate results. To determine the average cob length, the number of cobs measured per plot is divided by the total of individual measurements.

### Cob girth (cm)

Measurement of maize cob girth was conducted following the harvest of mature maize cobs. After removing any extraneous leaves or husks from five randomly selected cobs per plot, each cob girth was individually measured using a flexible tape. The measurement was taken at the widest point around the circumference of the cob, ensuring the tape was snug but not overly tight.

### Mean grain cob yield (t ha$^{-1}$)

Mean grain cob yield refers to the average amount of grain produced per cob of maize. The mean grain cob yield was calculated whereby the total number of cobs harvested per plot was divided by the total weight of grain harvested from five randomly selected cobs per plot. This provides an average measure of the productivity of individual maize cobs in terms of grain yield.

### Cob yield (t ha$^{-1}$)

The cumulative weight of cobs harvested from each net plot across all pickings was initially measured in kilograms. Subsequently, this cumulative weight was converted to ($\mathbf{t\,ha^{-1}}$) to provide a standardized measure of the overall yield of cobs across the experimental plots.

### Test weight (100 kernel weight)

The test weight was estimated by specifically weighing the weight of 100 randomly selected grains from each plot. Using a precise weighing scale, we measured the total weight of these 100 grains, ensuring the scale was properly calibrated for accuracy. The recorded total weight of the 100 grains was then divided by 100 to calculate the average weight of a single grain. This average weight, expressed in grams, represented the test weight or 100 kernel weight of the maize samples.

### Stover yield (t ha$^{-1}$)

After the completion of pickings, the green fodder harvested from each net plot was bundled and weighed, with the measurements recorded in kilograms per plot (kg plot$^{-1}$). To standardize the data and facilitate comparison, these weights were converted into $\mathbf{t\,ha^{-1}}$. It's important to note that the husk obtained during the harvesting process was also included in the total fodder yield calculation, providing a comprehensive assessment of the overall green fodder productivity across the experimental plots.

### Statistical analysis

The results obtained from diverse parameters underwent statistical analysis employing established methodologies. Descriptive statistics, three-factor analysis, and analysis of variance (ANOVA) were conducted, followed by mean comparisons using Tukey's HSD test at a significance level of $p < 0.05$ utilizing the R software. This thorough statistical

evaluation facilitated a comprehensive examination of the data, enhancing insights into the relationships among various factors and their significance.

## RESULTS

### Pyrolyzed and unpyrolyzed residues affect maize growth and productivity

Plant height of maize crops treated with various amendments (A1 to A7) ranged from 168.22 to 293.87 cm (Table 2). The tallest plants were observed with amendment A3 (293.87 cm), followed by A6 (264.23 cm), while the shortest plants were associated with A1 (168.22 cm). Notably, amendments A3 and A5 showed significant differences compared to other treatments, while A2, A6, and A1, A4, and A7 were similar in their effects. Dry weights of maize under various amendments, both pyrolyzed and unpyrolyzed, are shown in Table 2. Among the treatments, the highest dry weight was recorded with amendment A4 (14.11 t ha$^{-1}$), followed by A3 (13.64 t ha$^{-1}$), while the lowest dry weight was observed with amendment A1 (7.37 t ha$^{-1}$). Dry weights for amendments A1 to A7 ranged from 7.37 to 14.11 t ha$^{-1}$. Significant differences were noted in dry weight among amendments A1, A2, A4, and A6, whereas amendments A3, A5, and A7 showed no significant differences in dry weight. The average number of cobs per plant ranged from 0.49 to 1.75. Amendment A6 exhibited the highest number of cobs per plant (1.77), followed closely by A3 (1.71), while A1 had the lowest (0.49). The order of mean cobs per plant was: A6 >A3 >A5 >A7 >A4 >A2 >A1. Notably, there were no significant differences in the number of cobs per plant between amendments A3, A5, and A6, or between A4 and A7. However, there were considerable variations in the number of cobs per plant between amendments A1 and A2. The number of kernels per cob ranged from 387.36 to 535.75. Amendment A3 had the most kernels per cob (535.75), followed by A6 (497.33), while A1 had the least (387.36). The order of kernels per cob from highest to lowest was A6 >A3 >A2 >A5 >A4 >A7 >A1. Notably, there were no significant differences in the number of kernels per cob between A2, A4, A5, and A7. However, there were notable differences between A1, A3, and A6. The SPAD values for maize with amendments A1 to A7 were 32.27, 55.55, 58.10, 31.94, 48.30, 52.08, and 35.61, respectively. These values ranged from 31.94 to 58.10, with A3 having the highest (58.10) and A4 the lowest (31.94). The order of SPAD values from highest to lowest was A3 >A2 >A6 >A5 >A7 >A1 >A4. Significant differences existed between A2, A3, A5, and A6, while A1, A4, and A7 were comparable. The cob length values for maize with amendments A1 to A7 were 16.48, 26.29, 27.36, 16.34, 23.23, 24.82, and 17.89 cm, respectively. These values ranged from 16.34 to 27.36 cm. Amendment A3 had the highest cob length (27.36 cm), followed by A2 (26.29 cm), and the lowest was A4 (16.34 cm). The order of cob length values was A3 >A2 >A6 >A5 >A7 >A1 >A4. Significant differences were found between A2, A3, A5, and A6, while A1, A4, and A7 were similar. Table 2 shows the cob girth of maize with various residues. Amendment A3 had the largest cob girth at 18.18 cm, followed by A2 at 17.65 cm, while the smallest was with A4 at 12.75 cm. The cob girth ranged from 12.82 to 18.18 cm across the different amendments. Significant differences were noted in cob girth for A2, A3, A5, and A6, while A1, A4, and A7 showed no significant
**Table 2  Effect of various amendments on maize growth and yield parameters.**

| Amendments | PH (cm) | DW (Mg ha$^{-1}$) | NCP | NKC | SPAD | CL (cm) | CG (cm) | MGCY (Mg ha$^{-1}$) | GY (Mg ha$^{-1}$) | TW (gm) | SY (Mg ha$^{-1}$) |
|---|---|---|---|---|---|---|---|---|---|---|---|
| A1 | 168.22 ± 2.53 d | 7.37 ± 0.14 d | 0.49 ± 0.02 d | 387.36 ± 6.55 d | 32.27 ± 0.51 d | 16.48 ± 0.22 d | 12.82 ± 0.11 d | 0.69 ± 0.04 e | 3.27 ± 0.11 d | 117.54 ± 2.61 d | 0.86 ± 0.04 d |
| A2 | 261.24 ± 3.68 b | 9.28 ± 0.21 c | 0.75 ± 0.04 c | 469.11 ± 6.63 c | 55.55 ± 1.21 ab | 26.29 ± 0.51 ab | 17.65 ± 0.26 ab | 1.15 ± 0.04 b | 3.57 ± 0.12 cd | 285.61 ± 9.4 ab | 2.27 ± 0.09 ab |
| A3 | 293.87 ± 3.34 a | 13.64 ± 0.35 ab | 1.71 ± 0.05 a | 535.75 ± 7.37 a | 58.10 ± 1.29 a | 27.36 ± 0.54 a | 18.18 ± 0.27 a | 1.40 ± 0.06 a | 4.78 ± 0.17 a | 305.42 ± 9.96 a | 2.50 ± 0.09 a |
| A4 | 172.22 ± 3.06 d | 14.11 ± 0.17 a | 1.16 ± 0.03 b | 447.05 ± 6.06 c | 31.94 ± 0.93 d | 16.34 ± 0.4 d | 12.75 ± 0.2 d | 0.98 ± 0.03 cd | 3.86 ± 0.13 bc | 119.97 ± 5.23 d | 0.86 ± 0.06 d |
| A5 | 220.76 ± 2.91 c | 13.16 ± 0.31 ab | 1.65 ± 0.04 a | 456.21 ± 6.53 c | 48.30 ± 1.02 c | 23.23 ± 0.43 c | 16.15 ± 0.22 c | 1.08 ± 0.03 bc | 4.03 ± 0.13 bc | 229.30 ± 7.92 c | 1.92 ± 0.07 c |
| A6 | 264.23 ± 2.11 b | 12.79 ± 0.36 b | 1.75 ± 0.03 a | 497.33 ± 7.03 b | 52.08 ± 1.03 bc | 24.82 ± 0.44 bc | 16.93 ± 0.22 bc | 1.21 ± 0.04 b | 4.26 ± 0.12 b | 258.62 ± 7.97 bc | 2.16 ± 0.07 bc |
| A7 | 176.97 ± 2.43 d | 12.99 ± 0.33 ab | 1.27 ± 0.03 b | 443.76 ± 5.34 c | 35.61 ± 0.89 d | 17.89 ± 0.38 d | 13.51 ± 0.19 d | 0.86 ± 0.04 d | 3.59 ± 0.13 cd | 139.49 ± 5.45 d | 1.09 ± 0.06 d |
| CD (Amendment) | 3.16 | 0.544 | 0.03 | 4.99 | 0.41 | 0.17 | 0.07 | 0.086 | 0.193 | 5.42 | 0.078 |
| SE (m) (Amendment) | 1.60 | 0.276 | 0.01 | 2.53 | 0.21 | 0.09 | 0.04 | 0.044 | 0.093 | 2.75 | 0.04 |

Notes.

PH, Plant height (cm); DW, Dry weight (Mg ha$^{-1}$); NCP, Number cobs plant$^{-1}$; NKC, Number of kernels cob$^{-1}$; SPAD, soil plant analysis development; CL, Cob length (cm); CG, Cob girth (cm); MGCY, Mean grain cob yield (Mg ha$^{-1}$); GY, Grain yield (Mg ha$^{-1}$); TW, Test weight (gm); SY, Stover yield (Mg ha$^{-1}$).

Different letters (a–d) indicate the significant differences according to Tukey's HSD test ($p \leq 0.05$); CD: Critical difference; SE (m): Standard error mean.

variation. The cob girth mean values ranked as follows: A3 >A2 >A6 >A5 >A7 >A1 >A4. Table 2 outlines the mean grain cob yield values for maize crops, ranging from 0.69 to 1.40 t ha$^{-1}$. Amendment A3 had the highest yield at 1.40 t ha$^{-1}$, followed by A6 at 1.21 Mg ha-1, while A1 had the lowest yield at 0.69 t ha$^{-1}$. The order of yield was A3 >A6 >A2 >A5 >A4 >A7 >A1. Notably, there was no significant difference in yield between A2 and A6, but substantial differences existed among A1, A3, A4, A5, and A7 amendments. In Table 2, the grain yield of maize with amendments A1, A2, A3, A4, A5, A6, and A7 ranged from 3.27 to 4.78 t ha$^{-1}$. Amendment A3 had the highest mean yield at 4.78 Mg ha$^{-1}$, followed by A6 at 4.26 Mg ha$^{-1}$, while A1 had the lowest at 3.27 t ha$^{-1}$. The order of mean yield was A3 >A6 >A5 >A4 >A7 >A2 >A1. Significant differences were observed between the A1, A3, and A6 amendments, while A2 and A7, as well as A4 and A5 amendments, showed comparable yields. The test weight mean values for maize ranged from 117.54 to 305.42 gm across different amendments. Amendment A3 had the highest test weight at 305.42 gm, followed by A2 at 285.61 gm, while A1 had the lowest at 117.54 gm. The order of test weight values was A3 >A2 >A6 >A5 >A7 >A4 >A1. Significant differences were observed between A2, A3, A5, and A6 amendments, while A1, A4, and A7 showed similar weights. Table 2 displays the stover yield of maize crops with different amendments. Amendment A3 yielded the highest stover at 2.50 t ha$^{-1}$, followed by A2 at 2.27 t ha$^{-1}$, while A4 had the lowest yield at 0.86 t ha$^{-1}$. The stover yield ranged from 0.86 to 2.50 t ha$^{-1}$ across amendments. Significant differences were.

## Comparative analysis for year of application of pyrolyzed and unpyrolyzed residues and its influences on maize growth and productivity

In the 1st year, maize plant heights ranged from 163.43 to 289.56 cm with amendments A1 to A7, while in the 2nd year, heights ranged from 173.01 to 298.18 cm. Amendment A3 consistently produced the tallest plants in both years, reaching 289.56 cm and 298.18 cm, followed by A6 with heights of 258.47 cm and 269.99 cm, respectively. The shortest plants were associated with A1, measuring 163.43 cm and 173.01 cm. Plant heights increased

overall from the 1st to the 2nd year for all amendments. Significant differences in plant height were noted between amendments A3 and A5 in both years, while A2 and A6, and A1, A4, and A7 showed no significance. Table 3 displays the dry weight of maize crops in the 1st and 2nd years under different residues, both pyrolyzed and unpyrolyzed. In the 1st year, A4 had the highest dry weight (13.89 Mg ha$^{-1}$), followed by A3 (13.16 Mg ha$^{-1}$), while A1 had the lowest (6.96 Mg ha$^{-1}$). In the 2nd year, A4 still topped with the highest dry weight (14.33 Mg ha-1), followed by A3 (14.12 Mg ha$^{-1}$), and A1 remained the lowest (7.78 Mg ha$^{-1}$). Throughout both years, A1 consistently showed the lowest dry weight and A4 the highest. Dry weight mean values ranged from 6.96 to 13.89 Mg ha$^{-1}$ in the 1st year and from 7.78 to 14.33 Mg ha$^{-1}$ in the 2nd year. Dry weight mean values increased for all amendments in the 2nd year compared to the 1st year. Significant differences in dry weight were observed between A1 and A2 amendments in both years, while A1 = A2, and A3 = A4 = A5 = A6 = A7 showed no significant differences in dry weight. In maize, the number of cobs per plant ranged from 0.45 to 1.71 in the 1st year and from 0.54 to 1.78 in the 2nd year. For both years, each amendment fared as A1: 0.45 and 0.54 cobs per plant, A2: 0.73 and 0.78, A3: 1.67 and 1.75, A4: 1.12 and 1.21, A5: 1.62 and 1.69, A6: 1.71 and 1.78, and A7: 1.23 and 1.31 (Table 3). The 2nd year consistently saw more cobs per plant across all amendments. In both years, A6 topped with the most cobs per plant (1.71 and 1.78), followed by A3 (1.67 and 1.75), while A1 had the fewest (0.45 and 0.54). Across both years, there were no significant differences in the number of cobs per plant between A3, A5, and A6 (*i.e.,* A3 = A5 = A6) or between A4 and A7 (*i.e.,* A4 = A7). However, there were significant differences between A1 and A2 in both years. In the 1st and 2nd years, the average number of kernels per cob ranged from 382.35 to 528.42 and 392.37 to 543.08, respectively. Specifically, for the 1st and 2nd years, amendment A1 had 382.35 and 392.37 kernels per cob, A2 had 464.46 and 473.75, A3 had 528.42 and 543.08, A4 had 442.44 and 451.67, A5 had 451.49 and 460.92, A6 had 492.64 and 502.01, and A7 had 438.78 and 448.75, respectively (Table 3). In both years, A3 had the most kernels per cob (528.42 and 543.08), followed by A6 (492.64 and 502.01), with A1 having the fewest (382.35 and 392.37). Across both years, the number of kernels per cob was higher in the 2nd year. In both years, there were no significant differences in the number of kernels per cob between A2 and A5 (*i.e.,* A2 = A5) or between A4 and A7 (*i.e.,* A4 = A7). However, there were significant differences in the number of kernels per cob between A1, A3, and A6. In the 1st year, the SPAD values in maize crops with amendments A1, A2, A3, A4, A5, A6, and A7 were 31.58, 54.40, 56.93, 29.66, 46.86, 50.64, and 33.47, respectively, while for the 2nd year, they were 32.96, 56.70, 59.27, 34.22, 49.74, 53.51, and 37.75 (Table 3). For the 1st year, SPAD values ranged from 29.66 to 56.93, and for the 2nd year, they ranged from 32.96 to 59.27. In both years, the highest SPAD value was with amendment A3 (56.93 and 59.27), followed by A2 (54.40 and 56.70). Conversely, the lowest SPAD value for the 1st year was with A4 (29.66), and for the 2nd year was with A1 (32.96). SPAD values increased in the 2nd year for all amendments compared to the 1st year. Significant SPAD differences were observed between A5 and A6 for both years, while A1, A4, and A7 were comparable (A1 = A4 = A7), as were A2 and A3 (A2 = A3). The maize cob length varied from 15.38 to 26.87 cm and 16.77 to 27.86 cm, respectively for both years, with amendments A1 to

A7 (Table 3). Amendment A3 had the longest cob length in both years (26.87 and 27.86 cm), followed by A2 (25.80 and 26.77 cm). The shortest cob length was with A4 in the 1st year (15.38 cm) and A1 in the 2nd year (16.77 cm). Cob length increased in the 2nd year for all amendments compared to the 1st year. Significant differences in cob length existed between A5 and A6 for both years, while A1, A4, and A7 were comparable (A1 = A4 = A7), as were A2 and A3 (A2 = A3). Table 3 shows maize cob girth for the 1st and 2nd year. In both years, A3 had the largest cob girth (17.94 and 18.42 cm), followed by A2 (17.41 and 17.89 cm). The smallest cob girth in the 1st year was with A4 (12.28 cm), and in the 2nd year, it was with A1 (12.96 cm). Cob girth for A1 to A7 in the 1st year ranged from 12.68 to 17.94 cm, and in the 2nd year, it ranged from 12.96 to 18.42 cm. Cob girth increased in the 2nd year. Significant differences in cob girth were found between A5 and A6 in both years. However, A1, A4, and A7 were similar (A1 = A4 = A7), as were A2 and A3 (A2 = A3). The grain cob yield in maize varied from 0.63 to 1.26 Mg ha$^{-1}$ in the 1st year and from 0.75 to 1.45 Mg ha$_{-1}$ in the 2nd year (Table 3). For amendments A1 to A7, grain cob yield in the 1st year ranged from 0.63 to 1.36 Mg ha$^{-1}$, and in the 2nd year, it ranged from 0.75 to 1.45 Mg ha$^{-1}$. Grain cob yield increased in the 2nd year. A3 had the highest yield (1.36 and 1.45 Mg ha$^{-1}$), followed by A6 (1.16 and 1.25 Mg ha$^{-1}$), while A1 had the lowest (0.63 and 0.75 Mg ha$^{-1}$) in both years. Grain cob yield was similar between A4 and A5 (A4 = A5) but differed significantly among A1, A2, A3, A6, and A7. In the maize crop, grain yield with amendments A1 to A7 ranged from 3.15 to 4.64 Mg ha$^{-1}$ in the 1st year and from 3.40 to 4.92 Mg ha$^{-1}$ in the 2nd year (Table 3). Amendment A3 had the highest yield (4.64 and 4.92 Mg ha$^{-1}$), followed by A6 (4.13 and 4.40 Mg ha$^{-1}$), with A1 having the lowest (3.15 and 3.40 Mg ha$^{-1}$) in both years. Yield differed significantly between A1, A3, A5, and A6, but A2, A4, and A7 (A1 = A4 = A7) were comparable. Test weight mean values ranged from 116.02 to 296.32 gm in the 1st year and from 119.06 to 312.52 gm in the 2nd year for amendments A1 to A7 (Table 3). In both years, test weight values ranged from 116.02 to 296.32 gm and 119.06 to 314.52 gm, respectively. A3 had the highest values (296.32 and 314.52 gm), followed by A2 (276.67 and 294.56 gm), while A1 had the lowest (116.02 and 119.06 gm) in both years. Test weight increased in the 2nd year. Significant differences were observed with A5 and A6, but A1, A4, and A7 (A1 = A4 = A5) and A2 and A3 (A2 = A3) were statistically similar. Table 3 outlines the stover yield of maize under various amendments over two years. The highest stover yield in both years was with A3 (24.20 and 25.93 Mg ha$^{-1}$), followed by A2 (20.77 and 24.55 Mg ha$^{-1}$). However, the lowest yield in the 1st year was with A4 (7.04 Mg ha$^{-1}$), and in the 2nd year was with A1 (9.18 Mg ha$^{-1}$). Stover yield ranged from 7.04 to 24.20 Mg ha$^{-1}$ in the 1st year and 9.18 to 25.93 Mg ha$^{-1}$ in the 2nd year. The 2nd-year yield increased compared to the 1st year. In the 1st year, significant differences were observed with A3 and A5, while A1, A4, and A7 (A1 = A4 = A7) and A2, A6 (A2 = A6) showed no significant difference. In the 2nd year, A5 and A6 had significant differences, while A1, A4, and A7 (A1 = A4 = A7) and A2, A3 (A2 = A3) did not.
**Table 3  Effect of year and applied amendments on maize growth and yield parameters.**

| Year | Amendment | PH (cm) | DW (Mg ha⁻¹) | NCP | NKC | SPAD | CL (cm) | CG (cm) | MGCY (Mg ha⁻¹) | GY (Mg ha⁻¹) | TW (gm) | SY (Mg ha⁻¹) |
|---|---|---|---|---|---|---|---|---|---|---|---|---|
| 1st Year | A1 | 163.43 ± 3.33 d | 6.96 ± 0.17 b | 0.45 ± 0.02 d | 382.35 ± 9.11 d | 31.58 ± 0.7 c | 16.19 ± 0.3 c | 12.68 ± 0.15 c | 0.63 ± 0.05 d | 3.15 ± 0.14 c | 116.02 ± 3.33 c | 7.99 ± 0.43 c |
| | A2 | 256.12 ± 5.15 b | 8.77 ± 0.26 b | 0.73 ± 0.05 c | 464.46 ± 9.54 bc | 54.40 ± 1.71 a | 25.80 ± 0.72 a | 17.41 ± 0.36 a | 1.10 ± 0.05 b | 3.45 ± 0.17 bc | 276.67 ± 13.28 a | 20.77 ± 1.27 ab |
| | A3 | 289.56 ± 4.92 a | 13.16 ± 0.52 a | 1.67 ± 0.07 a | 528.42 ± 9.82 a | 56.93 ± 1.82 a | 26.87 ± 0.77 a | 17.94 ± 0.38 a | 1.36 ± 0.08 a | 4.64 ± 0.23 a | 296.32 ± 14.12 a | 24.20 ± 1.16 a |
| | A4 | 166.65 ± 4.32 d | 13.89 ± 0.23 a | 1.12 ± 0.04 b | 442.44 ± 8.53 c | 29.66 ± 1.21 c | 15.38 ± 0.51 c | 12.28 ± 0.25 c | 0.93 ± 0.04 bc | 3.75 ± 0.17 bc | 116.32 ± 6.48 c | 7.04 ± 0.78 c |
| | A5 | 215.02 ± 3.91 c | 13.10 ± 0.39 a | 1.62 ± 0.05 a | 451.49 ± 9.38 bc | 46.86 ± 1.42 b | 22.63 ± 0.6 b | 15.85 ± 0.3 b | 1.03 ± 0.04 bc | 3.89 ± 0.18 abc | 218.12 ± 11 b | 18.18 ± 0.92 b |
| | A6 | 258.47 ± 2.6 b | 12.28 ± 0.53 a | 1.71 ± 0.05 a | 492.64 ± 9.98 ab | 50.64 ± 1.43 ab | 24.22 ± 0.61 ab | 16.63 ± 0.3 ab | 1.16 ± 0.06 ab | 4.13 ± 0.18 ab | 247.44 ± 11.11 ab | 20.62 ± 0.93 ab |
| | A7 | 172.22 ± 3.22 d | 13.04 ± 0.33 a | 1.23 ± 0.04 b | 438.78 ± 7.47 c | 33.47 ± 1.17 c | 16.98 ± 0.5 c | 13.07 ± 0.25 c | 0.81 ± 0.05 cd | 3.47 ± 0.15 bc | 129.12 ± 6.35 c | 9.51 ± 0.76 c |
| 2nd Year | A1 | 173.01 ± 3.55 d | 7.78 ± 0.17 c | 0.54 ± 0.02 d | 392.37 ± 9.52 d | 32.96 ± 0.72 c | 16.77 ± 0.31 c | 12.96 ± 0.15 c | 0.75 ± 0.05 d | 3.4 ± 0.15 c | 119.06 ± 4.08 c | 9.18 ± 0.47 c |
| | A2 | 266.35 ± 5.12 b | 9.79 ± 0.29 b | 0.78 ± 0.06 c | 473.75 ± 9.33 bc | 56.70 ± 1.72 a | 26.77 ± 0.73 a | 17.89 ± 0.36 a | 1.21 ± 0.05 b | 3.68 ± 0.17 bc | 294.56 ± 13.34 a | 24.55 ± 1.12 a |
| | A3 | 298.18 ± 4.41 a | 14.12 ± 0.44 a | 1.75 ± 0.07 a | 543.08 ± 10.98 a | 59.27 ± 1.82 a | 27.86 ± 0.77 a | 18.42 ± 0.38 a | 1.45 ± 0.08 a | 4.92 ± 0.24 a | 314.52 ± 14.13 a | 25.93 ± 1.19 a |
| | A4 | 177.80 ± 4.03 d | 14.33 ± 0.24 a | 1.21 ± 0.04 b | 451.67 ± 8.7 c | 34.22 ± 1.22 c | 17.3 ± 0.52 c | 13.22 ± 0.26 c | 1.04 ± 0.04 bc | 3.98 ± 0.18 bc | 123.61 ± 8.3 c | 9.99 ± 0.79 c |
| | A5 | 226.50 ± 3.96 c | 13.22 ± 0.49 a | 1.69 ± 0.05 a | 460.92 ± 9.22 bc | 49.74 ± 1.43 b | 23.84 ± 0.6 b | 16.45 ± 0.3 b | 1.13 ± 0.05 bc | 4.16 ± 0.18 abc | 240.48 ± 11.06 b | 20.4 ± 0.93 b |
| | A6 | 269.99 ± 2.76 b | 13.29 ± 0.46 a | 1.78 ± 0.05 a | 502.01 ± 10.07 ab | 53.51 ± 1.43 ab | 25.43 ± 0.61 ab | 17.23 ± 0.3 ab | 1.25 ± 0.06 ab | 4.40 ± 0.17 ab | 269.8 ± 11.11 ab | 22.49 ± 0.93 ab |
| | A7 | 181.72 ± 3.35 d | 12.93 ± 0.57 a | 1.31 ± 0.03 b | 448.75 ± 7.66 c | 37.75 ± 1.16 c | 18.79 ± 0.49 c | 13.96 ± 0.24 c | 0.91 ± 0.05 cd | 3.71 ± 0.15 bc | 149.85 ± 8.33 c | 12.28 ± 0.75 c |
| CD (Year) | | 1.67 | 0.29 | 0.017 | 2.66 | 0.21 | 0.41 | 0.07 | 0.004 | 0.045 | 1.02 | 0.290 |
| CD (Amendment) | | 3.15 | 0.544 | 0.019 | 4.99 | 0.41 | 0.78 | 0.17 | 0.007 | 0.086 | 1.93 | 0.542 |
| CD (Year*Amendment) | | 4.48 | 0.771 | 0.03 | 7.08 | 0.61 | 1.12 | 0.25 | 0.011 | 0.122 | 2.74 | 0.765 |
| SE (m) (Year) | | 0.85 | 0.147 | 0.009 | 1.35 | 0.11 | 0.21 | 0.04 | 0.002 | 0.023 | 0.52 | 0.147 |
| SE (m) (Amendment) | | 1.60 | 0.276 | 0.01 | 2.53 | 0.21 | 0.40 | 0.09 | 0.004 | 0.044 | 0.98 | 0.275 |
| SE (m) (Year*Amendment) | | 2.27 | 0.391 | 0.02 | 3.59 | 0.31 | 0.57 | 0.13 | 0.006 | 0.062 | 1.39 | 0.388 |

**Notes.**

PH, Plant height (cm); DW, Dry weight (Mg ha⁻¹); NCP, Number cobs plant⁻¹; NKC, Number of kernels cob⁻¹; SPAD, soil plant analysis development; CL, Cob length (cm); CG, Cob girth (cm); MGCY, Mean grain cob yield (Mg ha⁻¹); GY, Grain yield (Mg ha⁻¹); TW, Test weight (gm); SY, Stover yield (Mg ha⁻¹).

Different letters (a-d) indicate the significant differences according to Tukey's HSD test ($p \leq 0.05$); CD: Critical difference; SE (m): Standard error mean.

## Influence of different amendments and rate of application on maize growth and productivity

Figure 3A revealed plant height as affected by different amendments and application rates. Among the amendments, there were significant variations in plant height, as well as among the application rates. Across L, M, and H rates, A3 resulted in the tallest plants, with significant differences noted. Conversely, A1, A4, and A7 led to the shortest plants, showing no variation among them. Generally, the highest plant heights were achieved with the H application rate, followed by the M and L rates. Figure 3B illustrates how different residues and application rates affect dry weight. Dry weight varied significantly across different residues and application rates. For M and H application rates, A3 resulted in higher dry weight compared to other amendments. However, for L application rates, A4 showed greater dry weight compared to other residues. Conversely, the lowest dry weight across all application rates was observed with A1 and A2, which were statistically similar. Dry weight decreased in the following order: H >M >L application rates. The influence of different amendments and application rates on grain yield is shown in Fig. 3C. Grain yield varied significantly depending on the type of amendment and application rates. A3 application consistently led to the highest grain yield across L, M, and H application rates, while A1 resulted in the lowest grain yield. Grain yield was highest with the H application rate, followed by the M and L application rates. Figure 3D revealed stover yield as affected by various amendments and rates of application. Stover yield varied notably among the amendments and application rates. The A3 amendment consistently led to the highest stover yield across L, M, and H application rates, showing significant variation. Conversely, the lowest stover yield was associated with the A1 amendment at the L rate and the A4 amendment at the M and H rates. Stover yield was highest with the H application rate, followed by the M and L rates.

## Amendment variability and fertilizer regimes affect maize growth and productivity

Figure S1A revealed plant height affected by various amendments and fertilizer regimes. Plant height varied significantly among different fertilizer regimes and amendments. The tallest plants were observed with the A3 amendment when paired with the RDF. Conversely, the shortest plants were associated with the A1 and A7 amendments, regardless of fertilizer application. Overall, plants treated with the RDF were taller than those without fertilizer across all amendments. Statistically significant differences in plant height were noted between the two fertilizer regimes, with the RDF resulting in taller plants compared to N. Figure S1B depicts the effect of various amendments and fertilizer regimes on dry weight. Dry weight varied significantly depending on the fertilizer regime and the type of amendments applied. The highest dry weight was observed when using the A3 amendment with the RDF. Conversely, the lowest dry weight occurred when applying the A1 amendment with the RDF or N. Across different amendment types, maize crop dry weight was consistently higher when the RDF was applied compared to when N was used. The order of dry weight concerning the fertilizer regime was: RDF >N. Statistically significant differences in dry weight were observed between the two fertilizer regimes

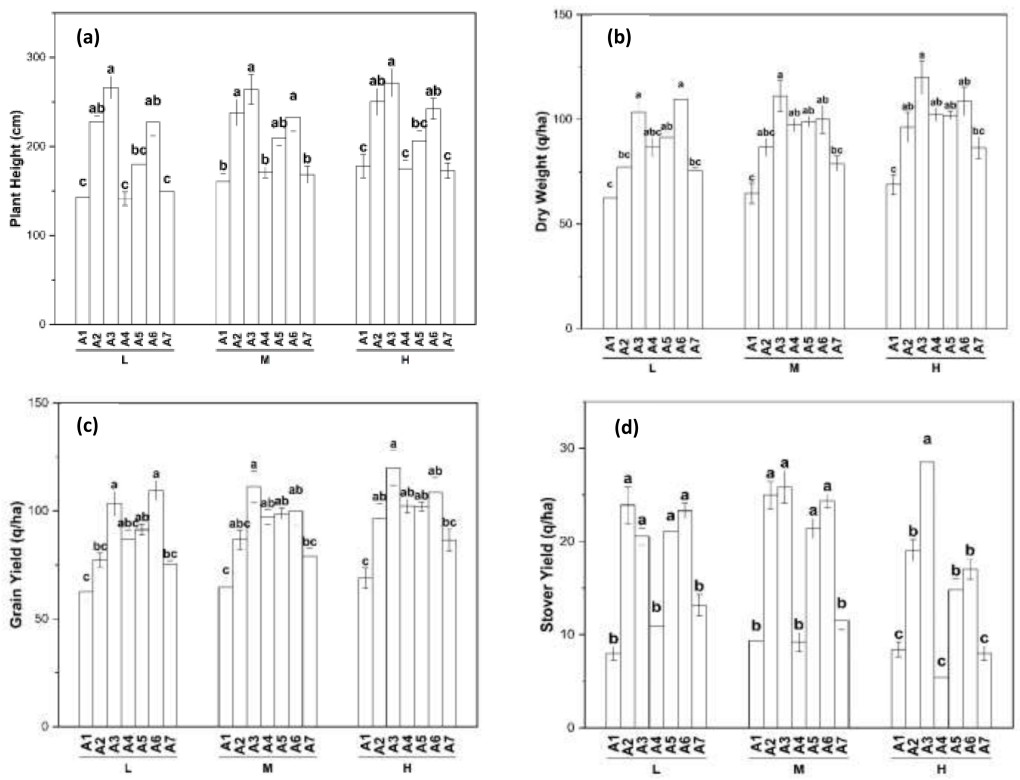

**Figure 3** Amendment and rate wise variation in plant height (A) dry weight (B) grain yield (C) and Stover yield (D).

across different amendments. However, the application of the A5 amendment showed no significant difference in dry weight whether paired with the RDF or N. The influence of different amendments and fertilizer regimes on grain yield is shown in Fig. S1C. Grain yield varied significantly only with the application of the A3 amendment, depending on the fertilizer regime. The A1, A2, A4, A5, A6, and A7 amendments did not show significant differences in grain yield between the two fertilizer regimes (N and RDF). Grain yield was higher when the RDF was applied compared to N, regardless of the amendment type. The order of grain yield concerning the fertilizer regime was: RDF >N. The A3 amendment resulted in the highest grain yield when applied with the RDF, while the A1 amendment produced the lowest grain yield with both fertilizer regimes. Figure S1D revealed stover yield affected by various amendments and fertilizer regimes. The stover yield varied significantly depending on the fertilizer regimes and the type of amendments. Applying the A3 amendment with the RDF resulted in the highest stover yield, while the lowest yield was observed with the A1 amendment, regardless of the fertilizer regime. Stover yield was generally higher when the RDF was applied compared to N. The order of stover yield concerning the fertilizer regime was: RDF >N. Significant differences in stover yield between the two fertilizer regimes were observed only with the A2 and A3 amendments,

while the A1, A4, A5, A6, and A7 amendments showed no significant differences in stover yield between the two fertilizer regimes.

## Interactive effect of the amendment, year, rate of application and fertilizer regimes on maize growth and productivity

Figure S2A illustrates an interactive means of the maize plant height affected by amendment type, year, rate of application and fertilizer regimes. The A3 amendment showed the highest average interaction with maize plant height, followed by A2, while A1 and A7 had the lowest values. Specifically, in the 2nd year, using A3 at a H rate with the RDF led to taller plants compared to other amendments, regardless of the application year, rate, or fertilizer use. Figure S2B represents the dry weight influenced by four factors: amendment type, year, rate of application, and fertilizer regimes. Among the amendments, A3 had the highest average interaction with dry maize weight, followed by A4, while A1 had the lowest. Specifically, in the 2nd year, using A3 at an H rate with the RDF led to greater dry weight compared to other amendments applied in the 1st year at M or L rates without fertilizer. Figure S2C depicts grain yield as affected by amendment type, year, rate of application, and fertilizer regimes. The highest mean grain yield was associated with the A3 amendment, followed by A4, and lowest with A1. Specifically, applying A3 in the 2nd year at a H rate with RDF led to the highest grain yield compared to other amendments applied at L rates or N in either the 1st or 2nd year. Figure S2D revealed stover yield as affected by amendment type, year, rate of application, and fertilizer regimes. A3 amendment application recorded the highest interactive mean with maize stover yield, followed by the A2, while the lowest was observed in the A1 and A7 amendment applications, respectively. The A3 amendment type, 2nd year of application, H rate of application and the inclusion of the RDF resulted in higher stover yield than the other amendment types in the 1st or 2nd year at a L or M rate with no fertilizer addition. Figure 4 represents scatterplot between yields and soil carbon.

## DISCUSSION

The research aimed to assess how pyrolyzed and un-pyrolyzed residues affect maize growth and productivity, examining parameters like plant height, dry weight, cobs per plant, kernels per cob, and SPAD value. The findings revealed significant effects of different amendments and application rates on these growth parameters. This study highlights the potential advantages of using certain residues to improve maize yield and productivity, confirming findings from previous studies that reported both positive and negative impacts (*Karer et al., 2013*; *Spokas et al., 2012*). The observed variations in plant heights among maize crops treated with different amendments (A1 to A7) reflect the intricate dynamics of soil management and nutrient supplementation in agricultural systems. The tallest plants, reaching 293.87 cm, were associated with amendment A3, followed by A6 at 264.23 cm, while the shortest plants, measuring 168.22 cm, were found with A1. These differences likely stem from various factors, including the nutrient composition of the amendments, soil structure and composition, microbial activity, pH levels, and root development (*Calamai et al., 2020*; *Hussain et al., 2017*; *Nardi, Schiavon & Francioso, 2021*). Amendments A3 and A6 may have provided optimal levels of essential nutrients and supported beneficial microbial

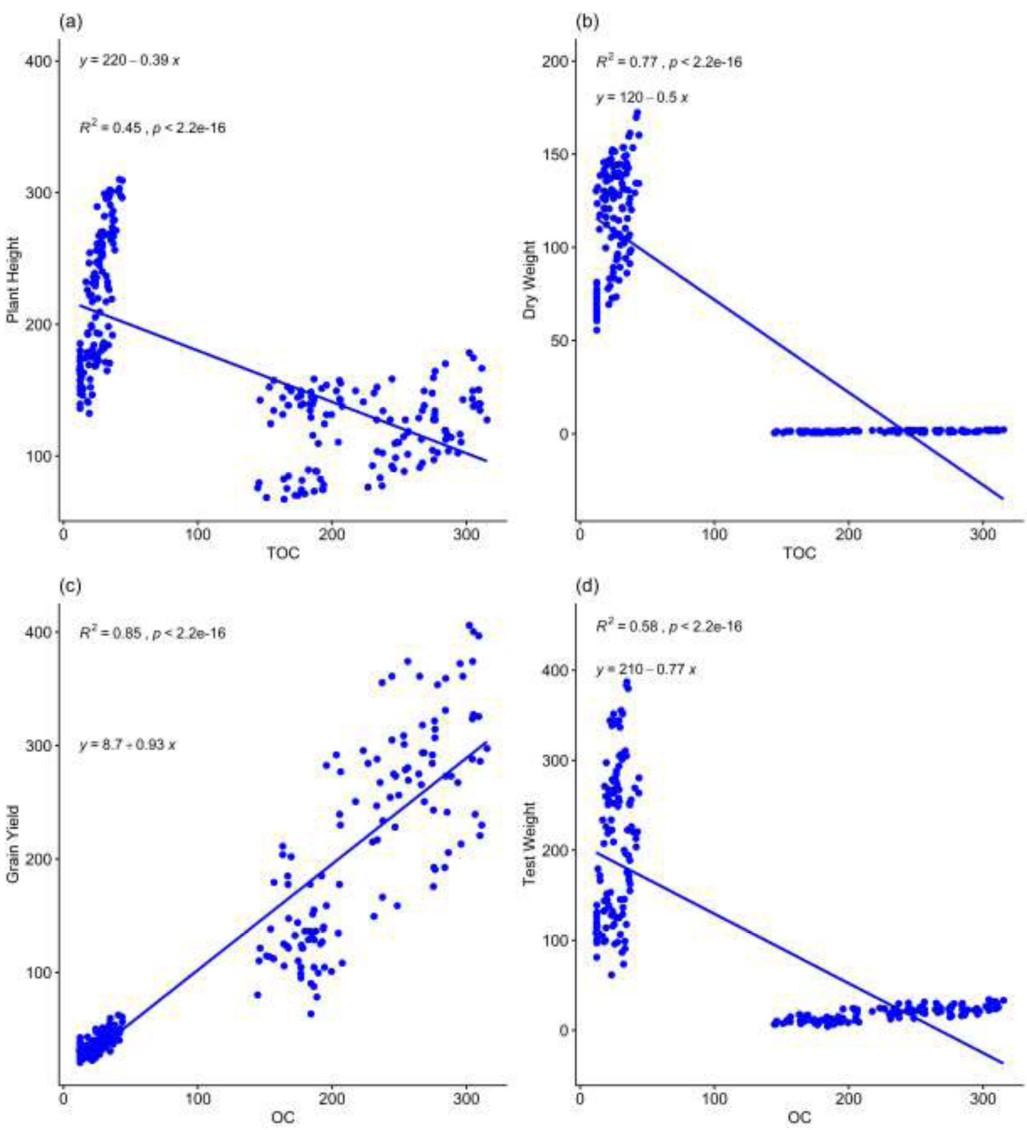

**Figure 4  Scatterplot between organic carbon and plant growth and yield parameters.**

communities, promoting robust vegetative growth and taller plant stature (*Bhadrecha, Singh & Dwibedi, 2023*; *de Bashan, Hernandez & Bashan, 2012*). In contrast, A1 might have lacked essential nutrients or contained substances inhibiting plant growth, leading to stunted heights. The significant differences observed between A3 and A5 compared to other treatments indicate distinct effects on plant height, likely attributable to their unique compositions and properties. Conversely, the similarities observed among A2, A6, as well as A1, A4, and A7, suggest comparable effects on plant height resulting from similar nutrient availability. Similar findings, such as significant variations in plant height among the different biochar amendments used, have been reported by *Calamai et al. (2020)*. The dry weights of maize under various amendments, both pyrolyzed and unpyrolyzed,

revealed significant differences in productivity across different treatments. Amendment A4 exhibited the highest dry weight, followed closely by A3, while the lowest was observed with A1. This variation in dry weight among treatments underscores the substantial impact of soil amendments on maize biomass accumulation and overall productivity (*Xiao et al., 2016*). The differences in dry weight among amendments can be attributed to several factors, including variations in nutrient availability, microbial activity, and organic matter content facilitated by the different amendments (*Gul et al., 2015*; *Zhang et al., 2016*). Amendments A4 and A3 may have provided optimal conditions for nutrient uptake and utilization, thereby promoting greater biomass accumulation in maize plants. These amendments might have contained essential nutrients, micronutrients, and organic matter in proportions conducive to robust plant growth and development (*Antonangelo, Sun & Zhang, 2021*). Conversely, the lowest dry weight observed with A1 could be attributed to inadequate nutrient supply or adverse effects on soil properties such as the capability of moisture retention, inhibiting maize growth and biomass production. The disparity in dry weights among amendments underscores the importance of selecting appropriate soil amendments tailored to specific soil and crop requirements to maximize productivity and yield potential. Observed differences in dry weight among amendments A1, A2, A4, and A6, with no significant differences noted among A3, A5, and A7, suggest distinct effects on maize productivity based on the type and composition of the amendments used (*Medyńska-Juraszek et al., 2021*). The findings of our study are in line with those reported by *Inal et al. (2015)*. *Butnan et al. (2015)* reported an increase ranging from 115% to 600% in maize dry weight compared to the control, achieved through the addition of biochar prepared at 350 °C at three different application rates (1%, 2%, and 4%) in loam sand (pH = 5.5) and silty clay loam (pH = 6) soils. As the temperature rises, the ash content of biochar increases, while the levels of hydrogen and oxygen decrease. Additionally, the biochar's aromatic structure intensifies, and its pH rises (*Mukome et al., 2013*; *Rajkovich et al., 2012*). Futhermore, the acidic aromatic carbon present on biochar's surface undergoes oxidation, resulting in the formation of plentiful functional groups (-OH, -COOH). This process enhances the soil cation adsorption capacity and boosts the soil's cation exchange capacity (CEC) (*Atkinson, Fitzgerald & Hipps, 2010*). The average number of cobs per plant, as observed across different treatments, ranged from 0.49 to 1.75, indicating notable variations in maize reproductive structures under the influence of various amendments. Each treatment exhibited distinct effects on the number of cobs per plant, with specific amendments demonstrating significant impacts on maize reproductive potential (*Thi et al., 2022*). Amendment A6 emerged as the most influential, displaying the highest number of cobs per plant, closely followed by A3, while A1 exhibited the lowest. The differences in the number of cobs per plant among treatments reflect the varying effects of amendments on maize reproductive development and yield. Amendments capable of providing optimal conditions for flowering, pollination, and cob formation tend to support higher numbers of cobs per plant (*Wolna-Maruwka et al., 2021*). Such differences may stem from variations in nutrient composition, biochar properties, or other factors influencing plant reproductive physiology and yield potential (*Abbas et al., 2021*). Utilizing biochar derived from woody shrubs in heavy clay soil not only enhances soil

physical properties and maize yield but also improves soil aeration, increases available water capacity, and decreases bulk density. Additionally, it promotes overall soil health and contributes to enhanced crop yield (*Obia et al., 2018*). The number of kernels per cob is a key indicator of maize yield and grain quality (*Mutungi et al., 2019*). Each treatment demonstrated distinct effects on kernel production per cob, highlighting that the type of amendment influences maize reproductive structures. Amendment A3 emerged with the highest number of kernels, followed closely by A6, while A1 exhibited the lowest. The differences in kernel numbers per cob among treatments underscore the varying impacts of soil amendments on maize reproductive development and yield. Amendments capable of providing optimal conditions for pollination, fertilization, and kernel development tend to support higher kernel numbers per cob (*Li et al., 2023*). Amendments A3 and A6 may have facilitated favourable soil conditions (*i.e.,* increased pH, improved soil structure and texture), nutrient availability, and physiological processes conducive to increased kernel production, leading to higher kernel numbers per cob compared to other treatments (*Das, Ghosh & Avasthe, 2017*; *Rácz et al., 2021*). The increase in kernels per cob implies that biochar may enhance the reproductive stages of maize, including flowering, pollination, and filling. We deduce that enhanced nutrient and water absorption capacities under biochar amendment likely play a significant role. The SPAD values, which serve as indicators of chlorophyll content and overall plant health, exhibited considerable variability among maize crops treated with different amendments (A1 to A7) (*Cortazar et al., 2015*; *Asai et al., 2009*). Amendment A3 recorded the highest SPAD value, while A4 had the lowest. Significant differences were observed between A2, A3, A5, and A6, indicating distinct effects on chlorophyll content and plant health under these treatments, while A1, A4, and A7 showed comparable SPAD values. The variations in SPAD values among treatments reflect the differential impacts of soil amendments on maize physiological processes and nutrient uptake (*Romdhane et al., 2021*). Amendments promoting optimal soil conditions, nutrient availability, and physiological functioning tend to result in higher SPAD values and healthier plants (*Agegnehu, Nelson & Bird, 2016*; *Cong et al., 2023*). The cob length and girth values for maize treated with amendments A1 to A7 exhibited notable variations in cob size across treatments. Amendment A3 displayed the longest cob length and girth, followed by A2, while A4 showed the shortest. The observed variations in cob length and girth among treatments highlight the diverse impacts of different amendments on maize reproductive development and cob elongation. Amendments providing optimal conditions for nutrient uptake, water availability, and physiological processes tend to result in longer cob length and girth and enhanced reproductive output (*Thomas, 2021*). These differences suggest distinct effects of soil amendments on cob development and girth, reflecting variations in nutrient availability, soil moisture, and genetic factors influencing maize reproductive structures (*Ndubo, 2023*). The variations observed in grain cob yield, grain yield, test weight, and stover yield among maize crops treated with different amendments (A1 to A7) reflect the complex interplay of soil fertility, nutrient availability, plant physiology, and environmental factors. Several key factors contribute to the differences in yield and quality across treatments. Different amendment composition directly influence nutrient availability, with A3 and A6 likely providing optimal levels of essential nutrients, promoting vigorous plant growth and

higher yields compared to amendments with lower nutrient availability, such as A1 (*Wani et al., 2023a*). Amendments play a crucial role in improving soil health and structure, with A3 and A6 potentially enhancing soil physical properties, water retention, and aeration, thereby facilitating better root growth and nutrient uptake compared to less effective amendments (*Xie et al., 2022*). Various biochar based amendments can influence various physiological processes within the plant, such as photosynthesis and nutrient assimilation, with A3 and A6 potentially enhancing these processes, leading to increased biomass production and higher yields compared to other treatments (*Hou et al., 2022*). The liming effect has been considered one of the mechanisms that elucidate the positive yield response to biochar amendment (*Pandit et al., 2018*). This assertion may hold particular significance in the acidic soils of tropical regions. Biochar inherently possesses a higher pH. When integrated into agricultural soil at tons per hectare, it elevates soil pH, consequently enhancing the availability of soil nutrients.

The highest plant height, dry weight, grain yield, and stover yield were noted with the H application rate, followed by the M and L application rates. These findings suggest that the choice of amendment type and application rates can substantially influence crop productivity. Biochar application enhanced grain yield, plant biomass, macronutrients, and micronutrients, resulting in a significant increase in grain yield, biomass, and macronutrient concentration in plants. Biochar, with its porous structure and large specific surface area, tends to decrease soil bulk density and increase total porosity, consequently enhancing root system development, improving plant nutrient uptake capacity, and stimulating maize growth and yield (*Ibrahim, Marie & Elfaki, 2021*). Also, biochar application contributes to soil structure enhancement by promoting the aggregation of soil mineral particles and increasing aggregate stability (*Cen et al., 2021*; *Zahed et al., 2022*). Since most biochar is derived from crop stalks, it is rich in nutrients. Numerous studies have confirmed that biochar application substantially elevates soil nutrient content, thus exerting a direct influence on crop growth (*Kamau et al., 2019*). *Ahmed et al. (2018)* reported that biochar did not affect maize yield in fully irrigated soils while it increased maize yield in reduced irrigated soils. They also found that biochar did not influence the maize yield at an amendment rate of 1%, while it increased maize yield at higher application rates (2% and 3%). Similarly, *Farooq et al. (2022)* and *Graef et al. (2018)* reported that in the sub-humid climate, biochar amendment enhanced maize yield solely under conditions of high application rates combined with low irrigation frequency. The findings highlight the critical role of amendment type and application rate in influencing maize productivity. The superiority of the H application rate in terms of plant height, dry weight, grain yield, and stover yield underscores the importance of proper nutrient management practices in maximizing crop performance. These results suggest that farmers can optimize their maize yields by selecting appropriate amendment types and applying them at the right rates. The benefits of biochar application on grain yield, biomass, and nutrient concentrations are consistent with previous studies. The porous structure and large surface area of biochar enhance soil properties, such as reduced bulk density and increased porosity, which are conducive to root growth and nutrient uptake. Additionally, the improvement in soil structure through enhanced aggregation and stability further supports the positive effects
of biochar on maize growth.The use of biochar in soil alters the circulation, retention, and conversion of nitrogen, leading to enhanced availability and reduced leaching of nitrogen in the soil (*Güereña et al., 2013*). This soil amendment, derived from renewable resources, can replace fossil-based soil improvers, thereby decreasing greenhouse gas emissions such as nitrous oxide ($N_2O$) and methane ($CH_4$) (*Zhang et al., 2019*).

Furthermore, findings underscore the significant impact of applying amendments and fertilizer regimes on maize growth parameters, particularly plant height and dry weight. Plant height varied significantly across different treatments and fertilizer applications, with the tallest plants observed with the application of the A3 amendment combined with RDF. Conversely, the lowest plant heights were recorded with the application of A1 and A7 amendments, either with or without fertilizer. This suggests that both the type of amendment and the presence of appropriate fertilizer play critical roles in influencing plant height. Similarly, the dry weight of maize crops exhibited substantial variation based on the type and application of amendments and fertilizers. The highest dry weight was achieved with the application of the A3 amendment and the RDF, indicating that this combination provided optimal conditions for biomass accumulation and plant growth. In contrast, the lowest dry weight was associated with the application of the A1 amendment, either with the RDF or without fertilizer altogether. The observed variations in plant height and dry weight of maize crops across different amendments and fertilizer regimes underscore the intricate interplay between soil management practices, nutrient availability, and plant growth responses. Biochar amendments, such as A3, when combined with the RDF, likely provided optimal levels of essential nutrients, promoting robust plant growth through enhanced nutrient availability. Additionally, amendments play a pivotal role in improving soil structure and health, facilitating better root development and nutrient absorption, particularly under treatments like A3 (*Wani et al., 2023b*). The interaction between amendments and fertilizers can result in synergistic effects on plant growth, with certain combinations promoting specific physiological processes within the plant and enhancing growth and productivity (*Rahman et al., 2021*). The application of the suitable amendment along with the RDF can notably increase the yield of maize crop. Several meta-analyses indicate that combining fertilizers with biochar leads to a substantial enhancement in maize yields across a range of agroecological settings (*Kumar et al., 2022*; *Singh et al., 2024*). Also, it is inferred that the application of the A3 amendment during the 2nd year, at a H application rate, in conjunction with the RDF, yielded the most pronounced positive effects on maize growth and yield attributes. *Naz et al. (2023)* revealed that when paired with micro-dosing of fertilizer, the impact on yield from biochar was significantly amplified (by 170%) compared to the control group. *Zahed et al. (2022)* similarly observed that when biochar was applied in combination with compost and chemical fertilizer, maize yield improved compared to the application of compost or chemical fertilizer alone. Consistent with our hypothesis, the introduction of biochar amendments increased maize grain production during the 2nd year following its application. This finding aligns with outcomes from prior research. However, *Niu et al. (2018)* observed no significant impact of biochar addition on maize yield in sandy loam soil within central China. These results portray the importance of employing suitable amendments and fertilizer regimes

to augment maize yield and productivity. Moreover, practices such as residue application, biochar utilization, and appropriate fertilizer application profoundly impact soil quality, C sequestration, and crop yields (*Nguyen et al., 2016*). The surface of biochar contains easily decomposable carbon and nitrogen sources, which are advantageous for bacterial decomposition. Bacteria can adhere to the biochar surface, rendering them less prone to leaching in the soil, thus increasing bacterial populations in soils (*Pietikäinen, Kiikkilä & Fritze, 2000*). The porosity and surface characteristics of biochar create a favorable environment for soil microbial growth and reproduction, reducing competition among microorganisms and safeguarding beneficial soil microbes, particularly root fungi, thereby enhancing their reproduction and activity (*Warnock et al., 2010*). The growth and decline of soil microbes influence the physical and chemical properties of soils, while changes in the soil microenvironment impact microbial growth. This, in turn, enhances soil fertility through microbial development and metabolism (*Ameloot et al., 2013*).

## CONCLUSIONS

Our study found that specific residues could enhance maize yield and productivity. The A3 amendment, applied at a high rate alongside recommended fertilizer, showed the best results, with the highest plant height, kernel count, SPAD value, cob length, cob girth, grain cob yield, grain yield, test weight, and stover yield. A4 and A6 amendments led to the highest maize dry weight and number of cobs per plant. Biochars produced at higher temperatures are of superior qualities as evidenced by functional characterization, which leads to improved soil physiochemical and biological properties, and hence enhancement of crop yields. Biochar produced at higher temperatures also helped in abiotic stress management and the performance of crops in adverse conditions, which gave it a yield advantage. Long-term use of biochar at high rates with recommended fertilizer significantly improved maize growth and production. There was a notable interactive effect among biochar types, rates, and fertilizer, highlighting biochar's impact on maize performance. The A3 amendment at a high rate with recommended fertilizer was the top performer for enhancing maize growth and productivity. Our findings support biochar as a sustainable agricultural amendment, emphasizing the importance of tailored application strategies for maximizing maize productivity and promoting environmental sustainability in agriculture.

## ACKNOWLEDGEMENTS

We acknowledge Division of Soil Science and Agricultural chemistry SKUAST Kashmir and NAHEP IDP SKUAST K for technical support.

### Funding

This work was supported by the Deanship of Scientific Research, King Saud University for funding through the Vice Deanship of Scientific Research Chairs; Research Chair of Prince

Sultan Bin Abdulaziz International Prize for Water. The funders had no role in study design, data collection and analysis, decision to publish, or preparation of the manuscript.

**Grant Disclosures**

The following grant information was disclosed by the authors:
The Deanship of Scientific Research, King Saud University for funding through the Vice Deanship of Scientific Research Chairs; Research Chair of Prince Sultan Bin Abdulaziz International Prize for Water.

**Competing Interests**

The authors declare there are no competing interests.

**Author Contributions**

- Owais Ali Wani conceived and designed the experiments, analyzed the data, authored or reviewed drafts of the article, and approved the final draft.
- Farida Akhter performed the experiments, analyzed the data, authored or reviewed drafts of the article, and approved the final draft.
- Shamal Shasang Kumar performed the experiments, analyzed the data, authored or reviewed drafts of the article, and approved the final draft.
- Raihana Habib Kanth performed the experiments, analyzed the data, authored or reviewed drafts of the article, and approved the final draft.
- Zahoor Ahmed Dar conceived and designed the experiments, performed the experiments, analyzed the data, authored or reviewed drafts of the article, and approved the final draft.
- Subhash Babu conceived and designed the experiments, analyzed the data, authored or reviewed drafts of the article, and approved the final draft.
- Nazir Hussain conceived and designed the experiments, prepared figures and/or tables, authored or reviewed drafts of the article, and approved the final draft.
- Syed Sheraz Mahdi conceived and designed the experiments, prepared figures and/or tables, authored or reviewed drafts of the article, and approved the final draft.
- Abed Alataway analyzed the data, prepared figures and/or tables, and approved the final draft.
- Ahmed Z. Dewidar analyzed the data, prepared figures and/or tables, and approved the final draft.
- Mohamed A. Mattar conceived and designed the experiments, performed the experiments, prepared figures and/or tables, and approved the final draft.

**Data Availability**

    The raw data are available in the Supplemental File.

**Supplemental Information**

Supplemental information for this article can be found online at http://dx.doi.org/10.7717/peerj.17513#supplemental-information.

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
