# Peer review of "Pyrolyzed and unpyrolyzed residues enhance maize yield under varying rates of application and fertilization regimes"

_PeerJ, doi:10.7717/peerj.17513_

## Round 0.1 · original submission · Major Revisions

Please focus on the findings and discuss what you have gained from them.

Reduce the length of the introduction, concentrating on topics relevant to your study.

The methodologies used are unclear, which could potentially lead to incorrect interpretation of the results. Therefore, please work on improving them.

Reviewer 1 ·

Basic reporting

Biochar has been widely used as a soil amendment in agricultural soils around the globe, but there is still a lack of information on the effect of different materials and different pyrolysis temperatures on maize yield. This study provides the appropriate basic data and is informative. However, this paper still has many flaws that need to be revised. For example, a large number of references need to be added to both the Introduction and Discussion sections, and the Results section needs to be further simplified. The Materials and Methods section is missing specific information on the experimental design and so on. In view of the valuable data, we need the authors to make significant revisions before considering publication.

Experimental design

Line 165, what is the soil type?
Line 170-172, missing specific experimental design. How were the plots laid out? What was the interval before the plots? How many each number of replicates? Information on the variety of maize and the corresponding field management is missing. This information needs to be added.

Validity of the findings

no comment

Additional comments

Specific comments are as follows:
Line 55, what is the full name of the SPAD. Abbreviations must be added in full the first time they appear.
Line 98-106, highlight the results that are relevant to this study and point out the shortcomings of the current study, cite your purpose.
Line 127-129, How do different pyrolysis temperatures and different application rates affect maize yield, cite literature. And explain how it is affected.
Line 213, the Results section needs major revision.
Compared to other sections, the results section is too long and needs to be further simplified to highlight the main results. For example, the same results could be combined into one sentence.
Line 558, the Discussion section likewise needs major revisions. First, the discussion section is too brief and much important information is not discussed. Second, too many results appear in the discussion section. For example, lines 560-567, which is the results section, does not need to be restated in the discussion, just a brief sentence or two. Finally, the Discussion section is missing some comparisons of the literature; are the results of this study the same as those of previous studies? Or are there differences. Further explanatory notes are needed.
Most importantly, what exactly is the reason for the results of this study? Is it because of the properties of the biochar under this treatment? Or the change in soil properties? Or is the difference due to different pyrolysis temperatures, and pyrolysis times.
line 602, the Conclusion section needs to highlight the most important results instead of repeating the abstract section. In short, the conclusion needs to be rewritten.

Reviewer 2 ·

Basic reporting

Any necessary corrections requested are marked in the manuscript.

Experimental design

There are major errors in article design. It needs to be rewritten.

Validity of the findings

The findings are very complex at this stage. It is not explained how the experiment was conducted and how the statistical data were obtained.

Additional comments

The authors analysed the changes in the yield parameters of organic wastes applied to soil at different pyrolysis temperatures and without any biochar. At first glance, the article seems to have major problems in editing and writing. Firstly, the experimental setup is not mentioned at all. Secondly, the authors mentioned the dose rates as low, medium and high in a way that is not found in the literature. This part should definitely be corrected. In future studies, researchers cannot know how much biochar will be added to the soil. As if biochar has never been analysed, no results are mentioned in the article. The figures have absolutely no readability and should be replaced with high resolution figures. The authors have repeated sentences in the conclusion and lost fluency. This section should be rewritten. In addition, Table 2, which is given as the average of the results of the 1st and 2nd years, is not necessary. The authors were careless in writing and submitting the manuscript because of meaningless explanations such as 1a in the table numbers. These need to be revised again. The authors have never hypothesised why they did this study, they should explain the hypothesis at the end of the introduction. The authors have never mentioned what kind of experimental design they set up, and accordingly, it is not clear which method was chosen for statistics.
Without the above explanations in the discussion section, it is not clear why biochar at 600 degrees gave good results.
The authors need to make major changes in the paper.

Annotated reviews are not available for download in order to protect the identity of reviewers who chose to remain anonymous.

---

## Round 0.2 · Major Revisions

The experimental design and methodology are confusing, potentially leading to misinterpretation of the results.

The discussion lacks depth. The authors should provide more specific interpretation of the findings and offer insightful analysis.

Please carefully address all reviewers' comments.

Reviewer 1 ·

Basic reporting

The author has revised the manuscript extensively, but there are still some issues that require further revision.

Experimental design

Lines 177-178, incorrect soil type description, state the specific soil type in this area based on one of the classification systems.
Lines 203-209, how was the biochar applied to the soil.

Validity of the findings

no comment

Additional comments

Lines 62-65, this conclusion is very common, need to summarize why this biochar type and application rate worked best, by affecting what properties of the soil or plant?
Results section, which turned out to be the same problem as before, is too long and discourages the reader from reading further. It is recommended that the authors utilize summarized results to state the specific findings of this study.
Conclusion section, further writing is needed to explain why these results occurred.
Figure 1, lack of latitude and longitude.
Additionally, the authors' response note does not provide any information, so next time, please indicate what was changed and how it was changed.

Reviewer 3 ·

Basic reporting

see attached file

Experimental design

it is unclear and confusing; therefore, difficult for other to reproduce similar/same type of experiment. Comments are in attached file.

Validity of the findings

see attached file.

Annotated reviews are not available for download in order to protect the identity of reviewers who chose to remain anonymous.

---

## Round 0.3 · Major Revisions

Abstract:
The abstract is excessively long and should be condensed while maintaining clarity and relevance. Additionally, it requires improvements in language and structure to enhance readability and impact.

Introduction:
The Introduction section (L89-179) contains redundant and repetitive information. Authors should streamline the content and clearly articulate the research hypothesis and expectations to provide a concise and focused introduction.
Italicize scientific names where appropriate.

References):
Ensure that all necessary references (L36-53, L108, L109 are included and properly cited throughout the text. Verify the format of references to meet the required style guidelines.

Materials and Methods:
Use SI units consistently throughout the Materials and Methods section to enhance clarity and standardization.
The methodology should be clear and concise, detailing methods, conditions, chemicals used, and steps. Avoid including unnecessary general introductions. Please improve this section.

Results:
Enhance the presentation of results, particularly in sections such as "Pyrolyzed and unpyrolyzed residues effects maize growth and productivity" by improving clarity and organization for better comprehension.

Discussion:
The discussion lacks specificity and should be revised to include a more in-depth analysis of the findings in relation to the practical applications of biochar and its effects on plant growth. Importantly, the authors should consider investigating changes in soil variables (e.g., pH, nutrients, minerals, microbiota) post-biochar application to provide a more comprehensive understanding of its impact on plant growth.

---

## Round 0.4 · accepted · Accept

The manuscript has much improved and can be accepted for publication.